# HSPA1A Can Alleviate CFA-Induced Inflammatory Pain by Modulating Macrophages

**DOI:** 10.3390/ijms26199591

**Published:** 2025-10-01

**Authors:** Wenjie Zhang, Xiaojun Xie, Xiaomin Xiong, Feiyu Chen

**Affiliations:** School of Basic Medical Sciences, Guangzhou Medical University, Guangzhou 510000, China; wenjiezhang0728@163.com (W.Z.); yu_zhou1212@163.com (X.X.); boboxiong1@163.com (X.X.)

**Keywords:** inflammatory pain, HSPA1A, macrophage, glycolysis, transcriptomics, proteomics

## Abstract

Current clinical approaches for managing inflammatory pain are frequently accompanied by adverse effects, significantly compromising patients’ quality of life. This study investigates the analgesic potential of Heat Shock Protein Family A Member 1A (HSPA1A) in alleviating Complete Freund’s Adjuvant (CFA)-induced inflammatory pain. The immunomodulatory mechanisms were elucidated through behavioral studies, flow cytometry, transcriptomics, proteomics, and cellular metabolic analyses. Findings indicate that HSPA1A mitigates CFA-induced mechanical allodynia, an effect independent of T or B lymphocytes and neutrophils but positively correlated with macrophage abundance. Transcriptomic RNA sequencing suggests involvement of inflammation-associated pathways. In vitro experiments demonstrate that HSPA1A suppresses the polarization of bone marrow-derived macrophages toward the pro-inflammatory M1 phenotype in an inflammatory model, with decreased mRNA expression of pro-inflammatory cytokines Interleukin-1β (Il1b) and Tumor Necrosis Factor (TNF). Macrophage metabolism undergoes reprogramming, characterized by reduced glycolysis and enhanced oxidative phosphorylation. Proteomic pathway analysis reveals suppression of pro-inflammatory and glycolytic proteins, coupled with upregulation of anti-inflammatory and tricarboxylic acid cycle-related proteins. In summary, HSPA1A likely exerts its analgesic effects by inhibiting glycolysis in macrophages, providing novel insights into inflammatory pain management and highlighting potential therapeutic targets for future clinical drug development with substantial translational potential.

## 1. Introduction

Inflammatory pain represents a significant challenge in clinical management, with its pathogenesis involving complex interplay between cellular and non-cellular immune mechanisms [1]. This pain is characterized by heightened excitability in primary afferent neurons, manifesting as spontaneous pain, hyperalgesia, and allodynia. Peripheral sensitization constitutes its core pathological process: on one hand, inflammatory mediators can elicit abnormal pain responses to normal stimuli; on the other hand, responsiveness to inflammatory stimuli is significantly amplified following tissue injury [2]. Inflammatory pain is classified into acute and chronic types. Acute inflammation serves as a physiological defense response to trauma or disease, activating nociceptors to trigger protective behaviors and promote tissue repair [3]. Chronic inflammation, however, persists pathologically, accompanied by intractable pain. Chemical mediators mediating tissue inflammation act on peripheral nociceptive nerve terminals, lowering neuronal excitation thresholds and increasing afferent firing rates, thereby inducing allodynia and hyperalgesia, respectively [4,5]. Animal models provide crucial tools for dissecting the pathophysiological mechanisms of chronic inflammatory pain and offer reliable data support for the screening and validation of candidate analgesic drugs [6,7].

Nociceptors, as somatosensory neurons, are widely distributed in the skin, cornea, urogenital tract, gastrointestinal tract, joints, bone, muscle, and deep visceral tissues [8,9]. Their peripheral nerve endings express molecular sensors such as Transient Receptor Potential (TRP) channels and G protein-coupled receptors (GPCRs), capable of detecting noxious inflammatory stimuli. These stimuli include reactive chemicals, extreme temperatures (heat and cold), and mechanical damage, as well as immune mediators like ATP, bradykinin, histamine, and cytokines [10,11]. Studies indicate that neuro-immune interactions in pain exhibit bidirectional regulatory properties [12,13]. Cytokines, lipids, growth factors, and other substances released by immune cells can act on peripheral nociceptors and central nervous system (CNS) neurons, mediating pain sensitization. Conversely, nociceptors can actively release neuropeptides from their peripheral terminals, modulating the functional states of innate and adaptive immune cells. Specifically, molecules including Interleukin-1β (Il1b), Tumor Necrosis Factor (TNF), Nerve Growth Factor (NGF), Prostaglandin E_2_ (PGE_2_), Leukotriene B_4_ (LTB_4_), and bradykinin can bind to their cognate receptors on nociceptive terminals, triggering neuronal firing. In summary, the immune system plays a key role in pain by releasing molecular mediators that sensitize nociceptive neurons. Tissue injury and inflammation are closely associated with increased pain perception. Nociceptor peripheral nerve endings possess receptors and ion channels that detect molecular mediators released during inflammation. Upon activation, action potentials are transduced to the nociceptor cell bodies within the dorsal root ganglion (DRG) [14]. Furthermore, immune mediators can enhance channel function by promoting membrane trafficking or upregulating their transcriptional expression. The combined effects of these immune-mediated pathways ultimately lower the response threshold of nociceptors to mechanical or thermal stimuli, resulting in heightened pain sensitivity [15].

Reports indicate that overexpression of Heat Shock Protein Family A Member 1A (HSPA1A) in joint tissues suppressed synovial inflammation, reduced chondrocyte apoptosis, and protected mice from osteoarthritis [16]. However, no study has definitively established the relationship between HSPA1A, immune cells, and inflammatory cytokines. Therefore, we will administer murine-derived HSPA1A protein to investigate its potential ameliorative effects on Complete Freund’s Adjuvant (CFA)-induced inflammatory pain and explore the underlying immune mechanisms.

## 2. Results

### 2.1. Effects of Different HSPA1A Doses on Inflammatory Pain Mouse Models

At 10 h post-CFA injection, paw withdrawal thresholds (PWTs) in C57BL/6 J mice decreased significantly to comparable levels, confirming successful establishment of CFA-induced inflammatory pain. Administration of 1 μg HSPA1A at 12 h and 20 h post-CFA significantly elevated PWTs (Figure 1A). Further investigation of HSPA1A dosing revealed that PWTs increased proportionally with higher HSPA1A doses, and the rate of PWT decline slowed as the administered dose increased (Figure 1B). Additionally, HSPA1A exerted consistent analgesic effects in both female and male mice, with no significant sex-based differences (Figure 1C).

### 2.2. Transcriptomic RNA Analysis of HSPA1A-Induced Regulatory Changes

Transcriptomic RNA analysis identified 1947 upregulated and 1817 downregulated genes in HSPA1A-treated mice versus controls (Figure 2A). Further analysis of pro-inflammatory genes revealed significant downregulation of IL1a, IL6, CXCL1, and others (Figure 2B). Gene Ontology (GO) enrichment analysis demonstrated that downregulated genes were significantly enriched in biological processes including vascular development, cytokine–cytokine interaction, the IL-17 signaling pathway, RAF-independent MAPK1/3 activation, and regulation of the ERK1 and ERK2 cascade (Figure 2C).

### 2.3. Impact of Different Immune Cells on HSPA1A-Mediated Analgesia

To determine whether immune cells influence HSPA1A efficacy, we compared inflammatory pain models in C57BL/6 J and NOD-SCID mice. NOD-SCID mice exhibit congenital immunodeficiency with minimal T and B cells (Figure 3A), yet HSPA1A retained its analgesic effect (Figure 3D). Neutrophil depletion in C57BL/6 J mice via anti-Gr1 antibody injection (Figure 3B) did not impair HSPA1A-mediated pain relief (Figure 3E). Conversely, macrophage depletion using clodronate liposomes (Figure 3C) significantly attenuated HSPA1A efficacy, resulting in lower PWTs versus controls (Figure 3F).

### 2.4. In Vitro Validation of HSPA1A Effects on Macrophage Polarization

Flow cytometry revealed leftward peak shifts for CD80, CD86, and MHC-II (M1 markers) in HSPA1A-treated BMDMs (LPS/IFN-γ) versus positive controls, while CD163 and CD206 (M2 markers) remained unchanged, indicating suppressed M1 polarization without affecting M2 differentiation (Figure 4A). qPCR confirmed downregulated *Il1b* and *Tnf* mRNA levels (Figure 4B). Adoptive transfer of BMDMs (LPS/IFN-γ, 1 × 10^4^ cells per mouse) into the paw elicited mechanical hyperalgesia, which was significantly attenuated by treating the cells with HSPA1A (Figure 4C). Metabolic analysis showed decreased ECAR and increased OCR in HSPA1A-treated BMDMs (LPS/IFN-γ) versus positive controls, indicating suppressed glycolysis and enhanced oxidative phosphorylation (Figure 4D).

### 2.5. Proteomic Profiling of DEPs and Enriched Pathways

Proteomic heatmaps identified differentially expressed proteins (DEPs) in HSPA1A-treated BMDMs, primarily involved in metabolic regulation and inflammation. Pro-inflammatory proteins (Cxcl2, CD80, NOS2, and TNF) were downregulated (Figure 5A), while anti-inflammatory proteins (Csf1r, CD36, Tgm2, and Bst2) were upregulated (Figure 5B). Glycolysis-related proteins (HK1, Gpi, Pfkl, and Pfkp) decreased (Figure 5C), whereas TCA cycle proteins (Suclg2, Suclg1, and Sdhb) increased (Figure 5D).

## 3. Discussion

Pain affects over 20% of the global population, imposing significant health and economic burdens. Effective pain management is crucial for sufferers. However, current pain assessment and treatment methods fall short of clinical needs. Thanks to advances in neuroscience and biotechnology, neuronal circuits and molecular mechanisms critically involved in pain modulation have been elucidated [17,18]. Inflammation causes pain, and pain is one of the primary signs of inflammation [19,20]. Clinical observations undoubtedly aid in treating inflammatory diseases, yet most fundamental discoveries regarding inflammatory mechanisms originate from animal models [21,22,23]. In vivo models help us clarify endogenous molecules involved in initiating and resolving inflammation and further improve our understanding of inflammatory pain. Moreover, animal models are essential for testing the efficacy and safety of new chemical entities with potential as novel anti-inflammatory analgesics. CFA is a conventional inducer for studying chronic inflammatory pain models in rodents. CFA injections induce production and release of various inflammatory mediators such as PGE2, nitric oxide, leukotriene B2, TNF-α, IL-2, and IL-17 [24,25]. These pro-inflammatory mediators can cause synovitis, polyarthritis, bone resorption, and periosteal bone proliferation and may lead to joint degeneration [26,27]. CFA is one of the superior models for studying inflammatory joint pain and is widely applied in research on skin inflammatory pain in mouse hind paws to investigate potential analgesic drugs. This study identified multiple inflammatory pain-related targets, all of which may provide new avenues for future pain treatment, offering theoretical foundations for clinically improving inflammatory pain management and addressing issues like opioid abuse and dependence.

Transcriptomic RNA analysis revealed that downregulated genes in HSPA1A-treated mice were enriched in pathways associated with the IL-17 signaling pathway, the MAPK signaling pathway, and the ERK signaling pathway. In temporomandibular joint osteoarthritis, M1 macrophage polarization has been shown to correlate with activation of the NF-κB/IL-17 pathway [28]. Meanwhile, RNA sequencing analysis in a model of LPS-pretreated cardiomyocyte-derived large extracellular vesicles (EVs) revealed significant enrichment of the MAPK pathway, suggesting its involvement in promoting M2-like polarization while suppressing the M1 phenotype [29]. Additionally, tumor necrosis factor (TNF)-stimulated gene-6 (TSG6) has been reported to promote tumor-associated macrophage polarization by activating the TGFβR1/Smad and MAPK/ERK pathways through direct interactions between CD44 and TGFβR1 or EGFR [30]. Whether HSPA1A modulates macrophage polarization to alleviate inflammatory pain through these or related signaling pathways warrants further investigation.

Macrophages are among the most important cells in the innate immune system. They transform into two distinct subtypes under different microenvironmental stimuli, exhibiting completely different molecular phenotypes and functional characteristics. Macrophages typically exist in two distinct subpopulations: (1) classically activated, or M1, macrophages, which are pro-inflammatory and polarized by LPS alone or in combination with Th1 cytokines, producing pro-inflammatory cytokines such as IL-1β, IL-6, IL-12, IL-23, and TNF-α; (2) alternatively activated, or M2, macrophages, which have anti-inflammatory and immunomodulatory effects, are polarized by Th2 cytokines like IL-4 and IL-13 and produce anti-inflammatory cytokines such as IL-10 and TGF-β [31,32]. Multiple studies indicate macrophages regulate inflammatory responses through different signaling pathways [33,34,35]. Expression levels of most inflammation-related proteins showed downward trends, while most anti-inflammatory proteins were upregulated, indicating HSPA1A may inhibit pro-inflammatory phenotypes in macrophages. As previously demonstrated, activation of the HSPA1A reduces NLRP3 inflammasome activity and the secretion of pro-inflammatory cytokines such as Il1b [36], supporting our observation. Metabolic reprogramming plays a key role in regulating macrophage function. Classically activated M1 macrophages favor glycolysis, producing lactate instead of metabolizing pyruvate to acetyl-CoA, and exhibit an interrupted TCA cycle at two points. M2 macrophages primarily utilize β-oxidation of fatty acids and oxidative phosphorylation to generate energy-rich molecules like ATP, participating in tissue repair and inflammation downregulation [37,38]. Song et al. found that Salvianolic Acid B attenuates liver fibrosis via suppression of glycolysis-dependent M1 macrophage polarization [39]. Whether metabolic reprogramming contributes to this HSPA1A-mediated polarization shift in inflammatory pain remains unclear. To explore this, we examined macrophage metabolism in vitro experiments and proteomics analysis and found that HSPA1A inhibits glycolysis while enhancing both oxidative phosphorylation and the tricarboxylic acid cycle, indicative of a metabolic reprogramming effect. These findings provide initial evidence that HSPA1A alleviates inflammation and pain through metabolic reprogramming of macrophages, offering new insights into the mechanisms behind its anti-inflammatory and analgesic actions.

Although our findings demonstrate that HSPA1A protein alleviates CFA-induced inflammatory pain by inhibiting glycolysis and suppressing M1 polarization of BMDMs, the specific binding receptors and activation states of related downstream signaling pathways remain unclear, which is a limitation of this study. In future studies, we will validate the expression of glycolysis-related proteins through in vivo and in vitro experiments and investigate glycolysis-associated signaling pathways in macrophages following HSPA1A treatment. Our study elucidated the mechanism through which HSPA1A regulates macrophage polarization by performing transcriptomic and proteomic sequencing of plantar tissues and BMDMs following HSPA1A treatment. Existing research indicates persistent increases in cytokines and chemokines in the central nervous system also promote chronic widespread pain affecting multiple body parts, and this modulation is associated with regulation of immune signals [40,41]. Therefore, to clarify the potential role of the CNS in the mechanism of HSPA1A action, future studies will investigate changes in the cerebrospinal fluid after HSPA1A administration. Furthermore, our findings demonstrate that HSPA1A downregulates the expression of the glycolytic enzymes HK1 and Pfkl in macrophages. Previous research has shown that demethylation of HK1 can modulate the glycolytic pathway [42], while phosphorylation of Pfkl, a rate-limiting enzyme in glycolysis, can activate metabolic reprogramming in macrophages [43]. Whether metabolic enzyme activity involved in metabolic reprogramming affects metabolic gene expression through epigenetic modifications such as methylation and phosphorylation was not deeply explored in this study. These limitations provide directions for future research.

In summary, we show that mouse-derived HSPA1A alleviates CFA-induced inflammatory pain by modulating macrophage polarization via glycolysis inhibition. However, the precise mechanism through which HSPA1A regulates glycolysis has not been fully elucidated, highlighting an important area for future investigation. Future studies will further explore molecular targets, expand disease models, and enhance clinical translation feasibility to bridge basic research and clinical applications.

## 4. Materials and Methods

### 4.1. Synthesis of HSPA1A Proteins

Recombinant murine HSPA1A protein was prepared as previously described [44]. Briefly, mouse HSPA1A cDNA and its N-terminal nucleotide-binding domain (NBD) and substrate-binding domain (SBD) fragments were amplified by PCR. These fragments were directionally cloned into the pET-22b+ vector using specific primers and restriction enzymes (NdeI and XhoI), transformed into E. coli DH5α, and positive clones were verified by PCR and sequencing. For recombinant protein production and purification, the validated plasmid was transformed into E. coli BL21(DE3). A single colony was cultured until OD reached 0.8–1.0, induced with IPTG for 14–16 h, and bacterial cells were collected by centrifugation. After lysis, the supernatant was incubated with cobalt–agarose beads to bind the target protein. Following washing, bound protein was eluted with imidazole-containing buffer and dialyzed into HEPES buffer using centrifugal filters.

### 4.2. Isolation and Culture of Bone Marrow-Derived Macrophages (BMDMs)

Femurs and tibiae from C57BL/6 J mice were dissected, adherent tissues removed, and bone marrow cells flushed with sterile PBS. After centrifugation, cells were resuspended in DMEM with 10% fetal bovine serum. Macrophage colony-stimulating factor (Peprotech, Cranbury, NJ, USA) was added to induce differentiation. Cells were cultured at 37 °C under 5% CO_2_ for 7–10 days, with the medium replaced every 2–3 days. Mature BMDMs were used for subsequent experiments.

### 4.3. Induction and Treatment of M1 Macrophages

Differentiated BMDMs were cultured in complete DMEM containing LPS (100 ng/mL, Beyotime, Haimen, China) and IFN-γ (20 ng/mL, Peprotech, USA) for 24 h to obtain M1 macrophages. For the experimental group, 100 ng/mL HSPA1A was added during this process; controls received an equal volume of solvent.

### 4.4. Establishment and Treatment of Inflammatory Pain Models

Thirty-nine C57BL/6 J mice (male, weighing 20 ± 2 g) and twelve NOD-SCID mice (male, weighing 20 ± 2 g) were purchased from Guangdong Zhiyuan Biomedical Technology Co., LTD (Shenzhen, China). Animals were housed conventionally with free access to food and water on a 12-hour light/dark cycle. All animal experimental procedures were approved by the Laboratory Animal Ethics Committee of Lai’an Technology (Guangzhou, China) Co., LTD (Approval number: G2025065, approval date: 7 August 2025) and were carried out in compliance with the ARRIVE 2.0 guidelines (https://arriveguidelines.org) accessed on 7 August 2025 regarding the care and use of laboratory animals. Into the plantar surface of the mouse hind paw was injected 10 μL of 1:1 diluted CFA (Sigma, Darmstadt, Germany). At 12 h and 20 h post-injection, mice received HSPA1A or an equal volume of saline (control). For neutrophil depletion, C57BL/6 J mice were intraperitoneally injected with anti-Gr1 antibody (49.5 μg dissolved in a total of 200 μL saline solution) 48 h and 24 h before CFA injection; controls received IgG antibody. For macrophage depletion, clodronate liposomes (300 μL) were administered intraperitoneally 72 h and 24 h before CFA; controls received empty liposomes.

### 4.5. Quantitative PCR (qPCR)

Total RNA was extracted using the AMV One-step RT-PCR Kit (Sangon, Shanghai, China) according to the manufacturer’s protocol. Reverse transcription was performed using the AMV First Strand cDNA Synthesis Kit (Sangon, CN). qPCR was conducted on a Roche LightCycler 96 system using 2 × SYBR Green Abstart PCR Mix (Sangon, CN), with primers listed in Table 1. Relative gene expression was calculated by the 2^−ΔΔCt^ method, normalized to GAPDH.

### 4.6. Measurement of Mechanical Hypersensitivity

Before behavioral experiments, mice were exposed to the testing environment for 1–2 h daily for 2 days without stimulation. All mice were acclimated to the testing environment. Experimenters were blinded to group assignments during testing. Each mouse was placed individually in a plexiglass chamber for 30 min habituation. Calibrated Semmes-Weinstein von Frey filaments (0.008, 0.02, 0.04, 0.07, 0.16, 0.4, 0.6, 1.0, and 1.4 g) were applied to the plantar surface of the inflamed paw for 3–4 s, with 5-minute intervals between stimuli. Filament forces were adjusted stepwise based on the mouse’s response. A positive response was defined as rapid withdrawal of the hind paw. After the first observed response change, five additional stimuli were applied. Paw withdrawal thresholds (PWTs) were assessed 8 times within 30 h post-CFA injection using the up–down von Frey testing paradigm. Baseline PWTs differed between NOD-SCID and C57BL/6 J mice; thus, NOD-SCID data were normalized.

### 4.7. Analysis of Cell Bioenergetics

Glycolysis and tricarboxylic acid (TCA) cycle activity in HSPA1A-treated BMDMs were measured using a Seahorse XFe96 Analyzer (Seahorse, La Verne, CA, USA). Extracellular acidification rate (ECAR) and oxygen consumption rate (OCR) were used to evaluate metabolic levels. As described [45], 25,000 BMDMs were seeded in XF-96 plates and treated. ECAR and OCR were measured using the XF-96 flux analyzer. Glycolytic parameters were calculated from ECAR changes after injections of glucose, oligomycin (OM), and 2-deoxyglucose (2-DG). Mitochondrial parameters were derived from OCR changes after injections of OM, FCCP, and rotenone + antimycin A (ROT + AA). ECAR and OCR data were normalized to DNA content measured by CyQuant.

### 4.8. Flow Cytometry

Plantar skin tissues were placed in cold PBS, minced, and digested with 0.2% collagenase at 37 °C for 30 min. Tissues were dispersed into single-cell suspensions, filtered through a cell strainer, and centrifuged at 1500 rpm for 5 min. The supernatant was discarded, and the pellet was resuspended in PBS. Cells were finally resuspended in PBS with 2% FBS at 1 × 10^6^ cells/mL. Cell suspensions were stained with CD45 (BioLegend, San Diego, CA, USA) antibody at 4 °C for 15 min protected from light. Antibodies against B cells (CD19, BioLegend, USA), NK cells (CD161c, BioLegend, USA), neutrophils (Ly6G, BioLegend, USA), and macrophages (CD64, BioLegend, USA) were added, mixed gently, and incubated at 4 °C for 30 min protected from light. After two PBS washes, cells were resuspended in 500 μL PBS and analyzed on a Beckman Cytoflex S flow cytometer (Beckman, Indianapolis, IN, USA). Data were analyzed using FlowJo (v10.4).

### 4.9. Transcriptomic Sequencing and Analysis

Plantar tissues from HSPA1A-treated and control mice were collected. Total RNA was isolated, treated with DNase, and purified using columns. RNA concentration was measured and adjusted. RNA quality was confirmed using a fragment analyzer. Qualified samples were used for library preparation: mRNA was enriched with oligo-dT beads, fragmented, and cDNA was synthesized. Adapters were ligated and PCR-amplified. Libraries were sequenced (paired-end) on an Illumina platform (https://www.illumina.com/, accessed on 7 August 2025). Raw data (FASTQ format) underwent base calling and quality control. Differentially expressed genes (DEGs) were identified (threshold: |log_2_FC| > 0.58, adjusted *p*-value < 0.05). DEG heatmaps were generated using the pheatmap package, and volcano plots were created with ggplot2. Downregulated DEGs were analyzed for GO enrichment using Metascape (Fisher’s exact test, *p* < 0.05).

### 4.10. Proteomic Sequencing and Analysis

Samples were prepared as above and submitted to APTBIO (Shanghai, China). Proteins were extracted using SDS-containing buffer and quantified. Labeled peptides were mixed, vacuum-dried, separated on a C18 trap column, and analyzed by LC-MS/MS. Peptides were identified by parsimony principles and filtered to a defined false discovery rate. Quantification positively identified peptides. Proteins with |log_2_FC| > 1 and an adjusted *p*-value < 0.05 were defined as differentially expressed proteins (DEPs). DEP heatmaps were generated using pheatmap.

### 4.11. Statistical Analysis

GraphPad Prism 9.5.1 was used for analysis. Data are expressed as mean ± SD. For statistical comparison between groups, two-group comparisons were made using two-tailed unpaired Student’s *t*-tests or Welch’s *t*-tests (for unequal variances). For comparisons of two groups over time, two-way ANOVA with the Sidak post hoc test was used. *p* < 0.05 was considered statistically significant.

## 5. Conclusions

HSPA1A protein alleviates CFA-induced inflammatory pain, and this effect specifically depends on macrophages. HSPA1A suppressed the M1 polarization of BMDM and reduced the secretion of the inflammatory cytokines IL-1β and TNF-α. This effect was associated with HSPA1A-induced metabolic reprogramming in macrophages, particularly through the inhibition of glycolysis (Figure 6). Our findings provide new experimental evidence clarifying the mechanism by which HSPA1A regulates macrophage polarization and highlight the potential of HSPA1A as a novel analgesic agent.

## Figures and Tables

**Figure 1 ijms-26-09591-f001:**
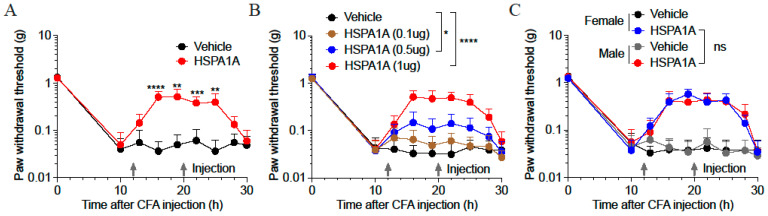
Effects of HSPA1A on paw PWTs in CFA-induced inflammatory pain mouse models. (**A**): PWTs in CFA-induced mice after HSPA1A injection (*n* = 9). (**B**): PWTs in CFA-induced mice after different HSPA1A doses (*n* = 7). (**C**): PWTs in female and male mice after HSPA1A injection (*n* = 7). * *p* < 0.05, ** *p* < 0.01, *** *p* < 0.001, **** *p* < 0.0001, ns: not significant (*p*  >  0.05).

**Figure 2 ijms-26-09591-f002:**
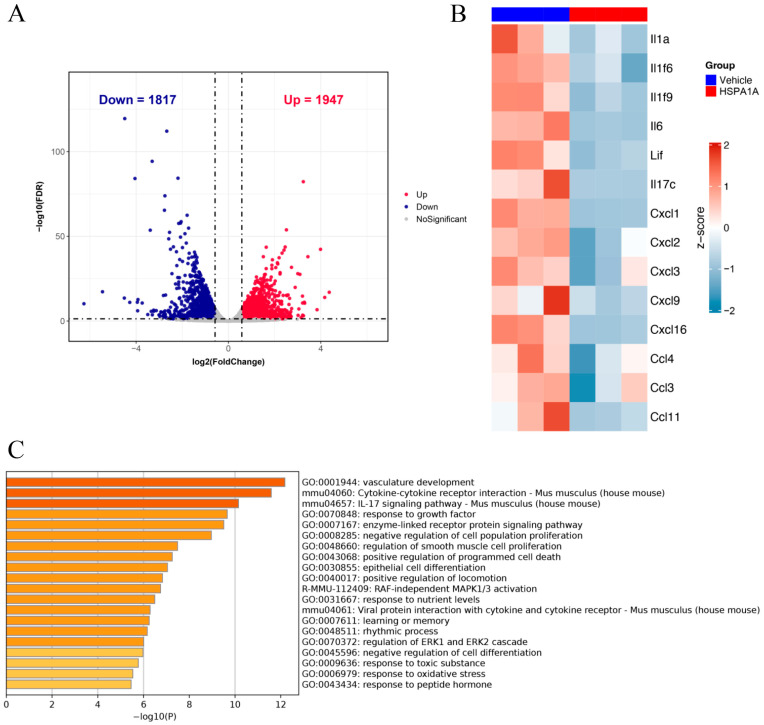
Transcriptomic RNA analysis. (**A**): Volcano plot of differentially expressed genes. (**B**): Heatmap showing expression changes of pro-inflammatory mediators in plantar tissues after HSPA1A injection. (**C**): GO enrichment bar chart displaying altered pathways.

**Figure 3 ijms-26-09591-f003:**
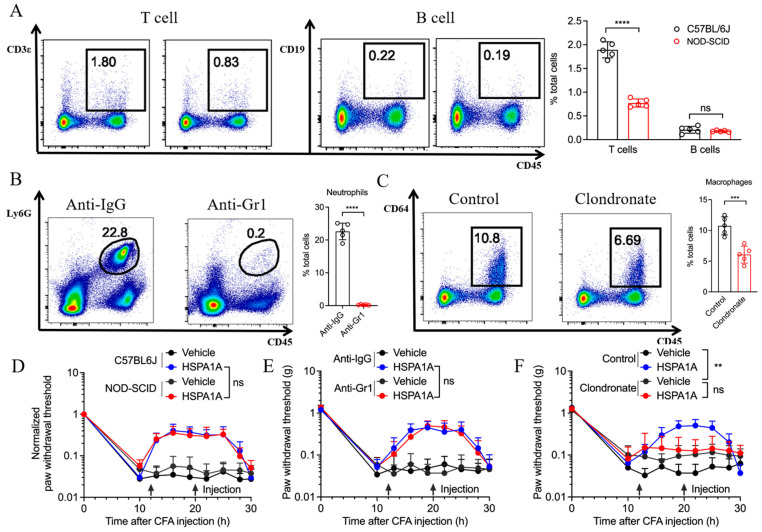
Effects of immune cell depletion on HSPA1A function. (**A**–**C**): Flow cytometry quantification of immune cells (*n* = 5) in plantar skin, (**A**): T/B cells, (**B**): Neutrophils, (**C**): Macrophages. (**D**–**F**): PWT measurements (*n* = 7). (**D**): T/B lymphocyte group, (**E**): Neutrophil group, (**F**): Macrophage group. ** *p* < 0.01, *** *p* < 0.001, **** *p* < 0.0001, ns: not significant (*p*  >  0.05).

**Figure 4 ijms-26-09591-f004:**
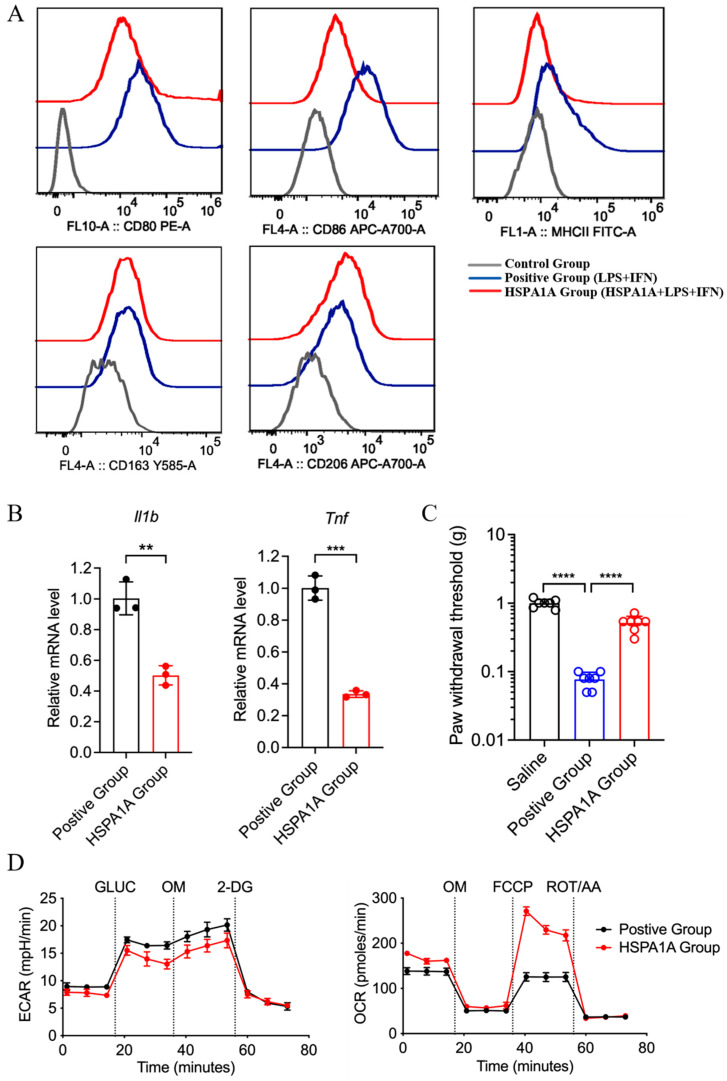
In vitro effects of HSPA1A on M1-polarized BMDMs. (**A**): Flow cytometry analysis of M1 (CD80, CD86, MHCII) and M2 (CD163, CD206) surface markers. (**B**): qPCR of Il1b and Tnf mRNA (*n* = 3). (**C**): PWT measurements (*n* = 7). (**D**): ECAR and OCR assessments (*n* = 6). ** *p* < 0.01, *** *p* < 0.001, **** *p* < 0.0001, ns: not significant (*p*  >  0.05).

**Figure 5 ijms-26-09591-f005:**
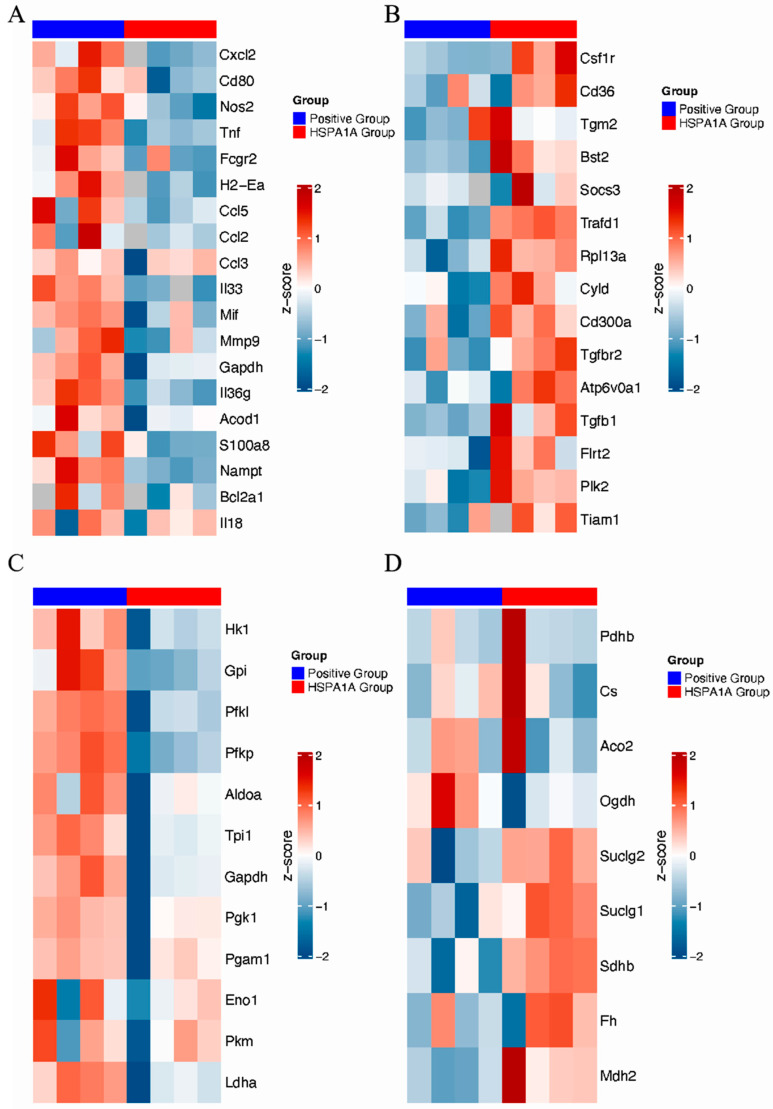
Proteomic analysis of HSPA1A-induced protein alterations in BMDMs. (**A**): Pro-inflammatory proteins. (**B**): Anti-inflammatory proteins. (**C**): Glycolysis-related proteins. (**D**): TCA cycle-related proteins. Protein expressions are shown as row-wise Z-scores (red, high; blue, low).

**Figure 6 ijms-26-09591-f006:**
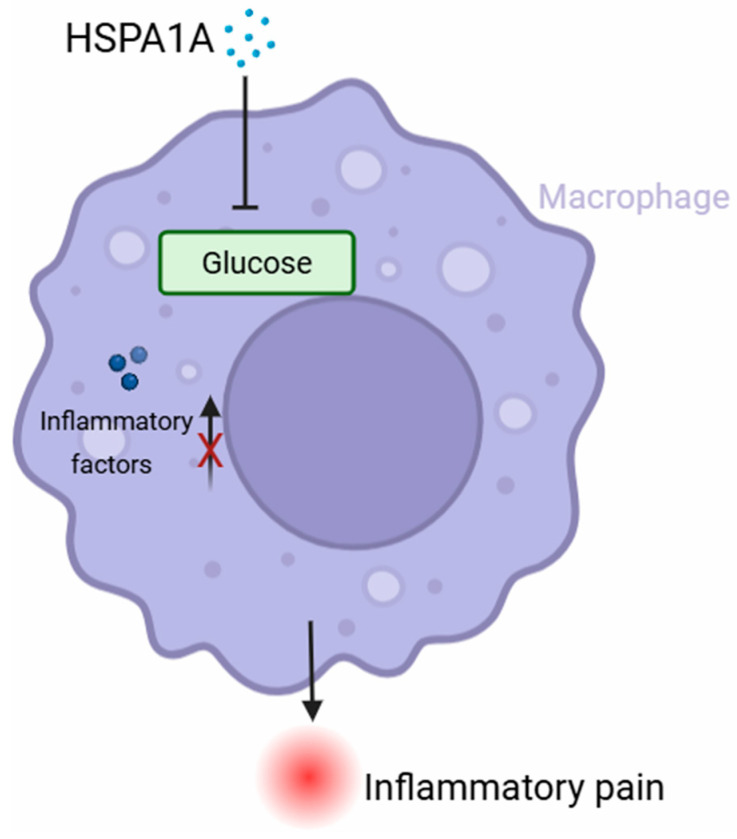
Schematic diagram. The present study demonstrated that HSPA1A significantly ameliorated CFA-induced inflammatory pain. This protective effect was achieved through inhibition of macrophage M1 polarization and glycolysis. HSPA1A inhibited glycolysis, thereby reducing macrophage M1 polarization and the release of inflammatory factors.

**Table 1 ijms-26-09591-t001:** Primer sequences.

Gene	Primer	
Il1b	F	5′- CCA CCT TTT GAC AGT GAT GAG -3′
	R	5′- CCA GGT CAA AGG TTT GGA AGC -3′
TNF-α	F	5′- GGT GCC TAT GTC TCA GCC TCT T -3′
	R	5′- GCC ATA GAA CTG ATG AGA GGG AG -3′
GAPDH	F	5′- GGG TGT GAA CCA CGA GAA AT -3′
	R	5′- CCT TCC ACA ATG CCA AAG TT -3′

## Data Availability

The transcriptomic sequencing data generated in this study have been deposited in the Gene Expression Omnibus (GEO) database under the accession number GSE304075, which is publicly accessible at https://www.ncbi.nlm.nih.gov/geo/, accessed on 7 August 2025. The proteomic datasets have been deposited in the iProX database with the accession number IPX0012817000 (available at https://www.iprox.cn/, accessed on 7 August 2025) and are also archived in the PRIDE database under the accession number PXD066909 (accessible at https://www.ebi.ac.uk/pride/, accessed on 7 August 2025). All other relevant data supporting the findings of this study are available from the corresponding author upon reasonable request.

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
