# Peer review of "HSPA1A Can Alleviate CFA-Induced Inflammatory Pain by Modulating Macrophages"

_ijms, 2025, doi:10.3390/ijms26199591_

Round 1
Reviewer 1 Report
Comments and Suggestions for Authors
My Comments are in attached word file.

Reviewer 2 Report
Comments and Suggestions for Authors
Please the comments below:
- Define HSPA1A
- Author shows previous study being conducted in rats where overexpressing HSPA1A exhibit decreased expression of inflammation-associated proteins, with reduced inflammatory cytokines. Have there been any previous similar studies using mouse models? If not, is this the first study in mouse administering HSPA1A to investigate its effects? Why wasn’t rat considered for this study if that is well established in the literature.
- Il1b, make it consistent throughout the manuscript.
- Line 295- “This aligns with literature reports that HSPA1A reduces inflammation levels.” Need appropriate citation to validate this.
- A schematic summary figure would be very useful to illustrate findings.
- Figure legend and significance value, please define asterisk value based on statistics and not include everything. Example Fig 1 ** is not presented on the figure but is included in the legend. This can confuse the reader.
